# Cryo-EM structure of the mechanically activated ion channel OSCA1.2

Sebastian Jojoa-Cruz[1†], Kei Saotome[1,2†], Swetha E Murthy[2], Che Chun Alex Tsui[1,3], Mark SP Sansom[3], Ardem Patapoutian[2*], Andrew B Ward[1*]

[1]Department of Integrative Structural and Computational Biology, The Scripps Research Institute, California, United States; [2]Department of Neuroscience, Dorris Neuroscience Center, Howard Hughes Medical Institute, The Scripps Research Institute, California, United States; [3]Department of Biochemistry, University of Oxford, Oxford, United Kingdom

**Abstract** Mechanically activated ion channels underlie touch, hearing, shear-stress sensing, and response to turgor pressure. OSCA/TMEM63s are a newly-identified family of eukaryotic mechanically activated ion channels opened by membrane tension. The structural underpinnings of OSCA/TMEM63 function are not explored. Here, we elucidate high resolution cryo-electron microscopy structures of OSCA1.2, revealing a dimeric architecture containing eleven transmembrane helices per subunit and surprising topological similarities to TMEM16 proteins. We locate the ion permeation pathway within each subunit by demonstrating that a conserved acidic residue is a determinant of channel conductance. Molecular dynamics simulations reveal membrane interactions, suggesting the role of lipids in OSCA1.2 gating. These results lay a foundation to decipher how the structural organization of OSCA/TMEM63 is suited for their roles as MA ion channels.

DOI: https://doi.org/10.7554/eLife.41845.001

*For correspondence:
ardem@scripps.edu (AP);
andrew@scripps.edu (ABW)

†These authors contributed equally to this work

Competing interests: The authors declare that no competing interests exist.

## Introduction

Mechanosensation allows organisms to detect and respond to external and internal mechanical forces (*Haswell et al., 2011*). Mechanical stimuli such as touch, gravity, and osmotic pressure are sensed by mechanically activated (MA) ion channels (*Haswell et al., 2011*; *Ranade et al., 2015*). Previously reported as putative ion channels sensitive to osmolality (*Hou et al., 2014*; *Yuan et al., 2014*; *Zhao et al., 2016*), a companion paper characterizes OSCA/TMEM63 family members as a novel family of MA ion channels conserved across eukaryotes (*Murthy et al., 2018*). Plants with mutant OSCA1 have impaired osmotic stress signaling (*Yuan et al., 2014*), implying that OSCAs function as sensors of osmotic-stress induced mechanical force *in vivo*. TMEM63 proteins are animal orthologues of OSCA that produce MA currents when expressed in naïve cells (*Murthy et al., 2018*), but their physiological roles remain undetermined. OSCA/TMEM63 are not homologous to other known MA ion channels, and no previous study has addressed their molecular structure. To facilitate a mechanistic understanding of how OSCAs sense force in plants, and to shed light on a newly identified family of eukaryotic MA ion channels, we conducted cryo-EM studies of OSCA1.2.

## Results

We purified *Arabidopsis thaliana* OSCA1.2 expressed in HEK293F cells and determined cryo-EM reconstructions in lipidic nanodiscs and Lauryl Maltose Neopentyl Glycol (LMNG) detergent micelle with cholesteryl hemisuccinate (CHS) at 3.1 and 3.5 Å resolution, respectively (*Figure 1A–D*,

**Figure 1.** Cryo-EM structure of OSCA1.2 and intersubunit cleft. Side (**A**, **B**), bottom (**C**), and top (**D**) views of EM density map of nanodisc-embedded OSCA1.2 dimer (sharpened map, blue and red). Nanodisc density is apparent at lower thresholds (unsharpened map, light grey). The intracellular domain (ICD) of each monomer is colored lighter than the transmembrane domain (TMD). Side (**E**, **F**), bottom (**G**), and top (**H**) views of OSCA1.2 dimer model. Coloring follows the same scheme used in density map. (**I**) Molecular dynamics (MD) simulations revealed the inter-subunit cleft to be occupied by phosphatidyl choline (PC) molecules arranged as a bilayer, shown in side (left) and top (right) views (also see *Figure 1—figure supplement 5A*) Lipid molecules shown are within 6 Å of residues 367–389 of TM3 (partial), 455–490 of TM5 and 616–685 of TM9–10 (also see *Video 2*).

DOI: https://doi.org/10.7554/eLife.41845.002

The following source data and figure supplements are available for figure 1:

**Source data 1.** Data collection, processing, model refinement, and validation.
DOI: https://doi.org/10.7554/eLife.41845.009
**Figure supplement 1.** Biochemistry and cryo-EM of OSCA1.2 in nanodiscs and LMNG.
DOI: https://doi.org/10.7554/eLife.41845.003
**Figure supplement 2.** Cryo-EM data processing flowchart.
DOI: https://doi.org/10.7554/eLife.41845.004
**Figure supplement 3.** Cryo-EM density and fit to model of OSCA1.2 in nanodisc.
DOI: https://doi.org/10.7554/eLife.41845.005
**Figure supplement 4.** Comparison between detergent and nanodisc-embedded OSCA1.2 and the intracellular dimeric interface.
DOI: https://doi.org/10.7554/eLife.41845.006

*Figure 1 continued*

**Figure supplement 5.** Protein-lipid interactions at the subunit interface and lipid and water molecules within the cytoplasmic region of the pore.
DOI: https://doi.org/10.7554/eLife.41845.007
**Figure supplement 6.** Root-mean-square deviation of protein during the atomistic simulation.
DOI: https://doi.org/10.7554/eLife.41845.008

*Figure 1—figure supplement 1*, *Figure 1—figure supplement 2*). The resulting density maps allowed building of 82% of the full-length OSCA1.2 sequence (*Figure 1—figure supplement 3*, *Figure 1—source data 1*). Structures of OSCA1.2 in nanodisc and LMNG were nearly identical (RMSD = 0.81 Å, *Figure 1—figure supplement 4A–C*). We refer to the nanodisc structure throughout unless otherwise noted because of superior resolution and map quality. Importantly, the purified channel reconstituted in liposomes retains MA ion channel activity (*Murthy et al., 2018*). Thus, our structures and interpretations correspond to that of a functional OSCA1.2 protein. Additionally, we performed MD simulations at both coarse-grained (CG) and atomistic (AT) levels to model the local membrane interactions of OSCA1.2 (*Figure 1—figure supplements 5–6*, *Video 1*).

The structure of OSCA1.2 reveals a trapezoid-shaped homodimer with a two-fold symmetry axis perpendicular to the membrane (*Figure 1A–H*). The dimer is 139 Å across at its widest dimension. The bulk of the protein lies within the membrane; the transmembrane (TM) domain (TMD) of each monomer contains eleven TM helices. Linkers between TM helices and a C-terminal region come together to form an intracellular domain (ICD) that extends ~31 Å below the membrane (*Figure 1A–H*). The ICD contributes the sole dimeric interface in OSCA1.2, with a buried surface area of 1,331 Å$^2$ (*Figure 1—figure supplement 4D*) (*Krissinel and Henrick, 2007*), only 4% of the total surface area of the dimer. Within the membrane, inter-subunit contacts are nonexistent. Instead, a cleft with minimum width of approximately 8 Å separates one subunit's TMD from the other (*Figure 1I*). Interestingly, AT-MD simulations of OSCA1.2 in a phospholipid (1-palmitoyl-2-oleoyl-*sn*-glycero-3-phosphocholine, POPC) bilayer, consistently place lipid molecules inside the cleft (*Figure 1I*, *Figure 1—figure supplement 5A*, *Video 2*), indicating a role for lipids in stabilizing the dimeric assembly.

*Figure 2A,B* show the topology and domain arrangement of an OSCA1.2 subunit. The N-terminus of OSCA1.2 faces the extracellular environment while the C-terminus is intracellular. The majority of the ICD is comprised of the second intracellular linker (IL2), which is over 150 residues and contains a 4-stranded antiparallel ß-sheet (IL2ß1-IL2ß4) with well-conserved sequences across the OSCA family (*Figure 2—figure supplements 1*, *2* and *3*), and four helices (IL2H1-IL2H4). Three additional helices, contributed by intracellular linkers 1 (IL1H2) and 4 (IL4H) and the C-terminus (CTH), constitute the rest of the intracellular domain. Other than short linkers, we did not observe structured extracellular domains, presumably due to flexibility.

As noted previously (*Hou et al., 2014*), a C-terminal region of OSCA1.2 (corresponding to TM4-TM9 in our structure) has loose homology to TMEM16 proteins, which are a family of Ca$^{2+}$-activated ion channels (*Caputo et al., 2008*; *Chen et al., 2010*; *Schroeder et al., 2008*; *Yang et al., 2008*) and lipid scramblases (*Malvezzi et al., 2013*; *Suzuki et al., 2013*; *Suzuki et al., 2010*) which, to our knowledge, have no mechanically gated activity describe so far. Strikingly, our structure reveals that the structural homology extends beyond the C-terminal region; ten of the eleven TM helices of OSCA1.2

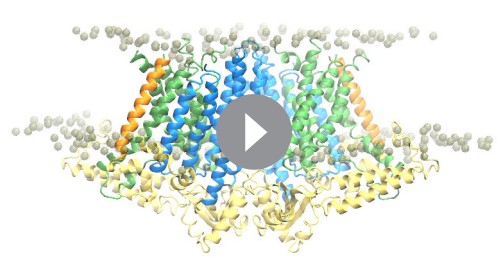

**Video 1.** Related to *Figure 2A* and *Figure 4A, B*. Atomistic simulation of OSCA1.2 (side view) in a POPC lipid environment. The system was first equilibrated for 1 µs in CG simulations before converting to atomistic detail. The entire AT simulation is 250 ns, with position restraints applied in the first 50 ns (see Materials and methods). This allows the lipid bilayer interactions with the protein to equilibrate. Phosphorus atoms of POPC lipids within 20 Å of the protein are shown as transparent tan spheres. The hook and IL1H1 regions remain inserted in the membrane. Lipid flip-flop (i.e. scramblase activity) was not observed over the course of the simulation.
DOI: https://doi.org/10.7554/eLife.41845.010

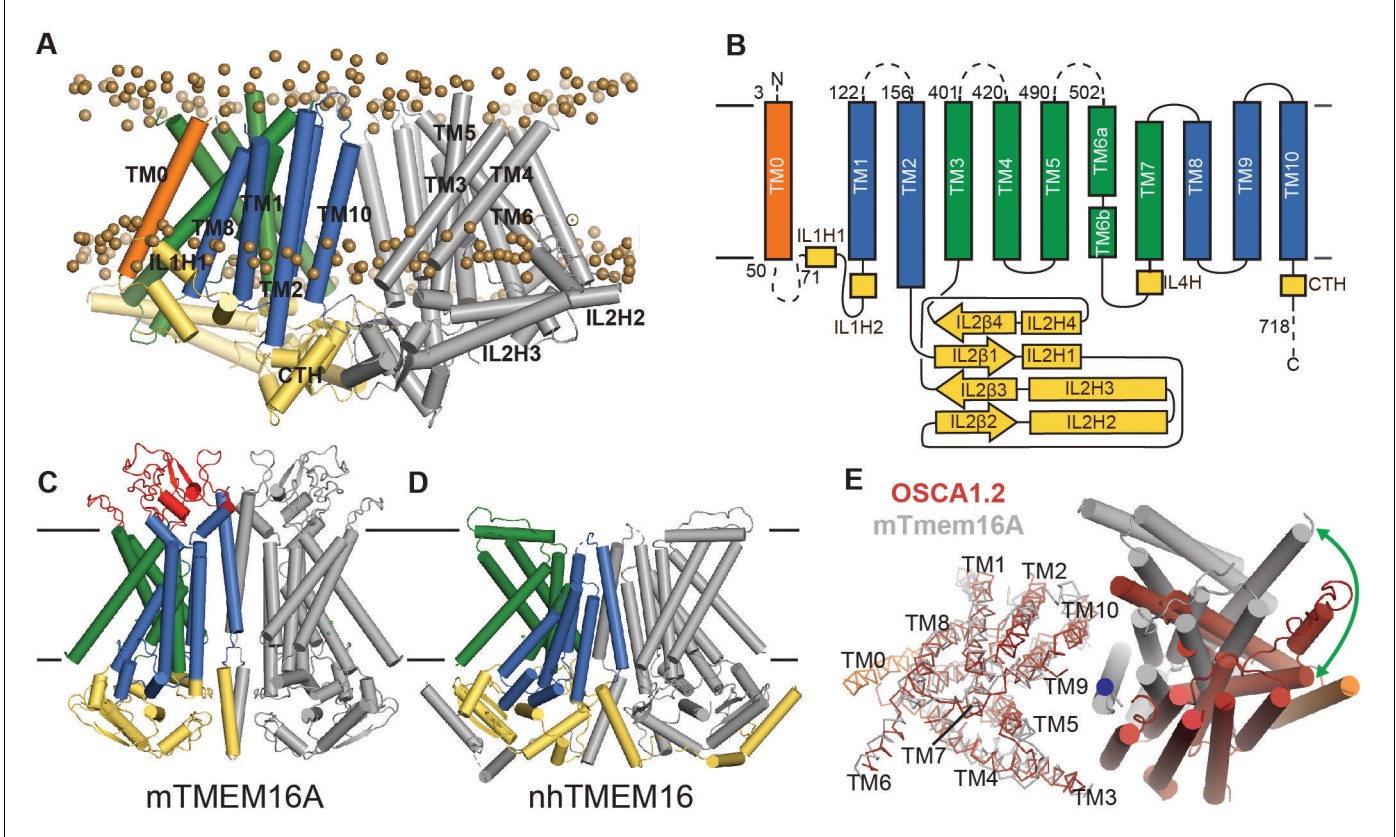

**Figure 2.** OSCA1.2 topology and comparison to TMEM16. (**A**) Structure of OSCA1.2 viewed from the membrane plane. One monomer is colored grey. The other monomer shows intracellular domain in yellow, TM0 in orange, pore lining TM helices in green (TM3–TM7) and TM helices not directly involved in the pore in blue. Phosphorus atoms of POPC molecules within 25 Å of the protein are shown as brown spheres (also see *Video 1*). (**B**) Topology diagram illustrating the secondary structure elements of OSCA1.2. Dashed lines represent missing residues in the model. A short coil divides TM6 into TM6a and TM6b. (**C, D**) Structures of mTMEM16A and nhTMEM16 (PDB: 6BGI and 4WIS, respectively) in the same view as (**A**). Extracellular domain of mTMEM16A is colored red. (**E**) Bottom view of the transmembrane domains of OSCA1.2 (red) and mTMEM16A (grey) aligned on one subunit (shown as ribbons). The extracellular and intracellular domains have been removed for clarity. The blue circle marks the symmetry axis of OSCA1.2. The green arrow marks the difference in position of TM6 in the non-aligned subunits (cylinders) between the two structures.

DOI: https://doi.org/10.7554/eLife.41845.012

The following figure supplements are available for figure 2:

**Figure supplement 1.** Protein sequence alignment of OSCA/TMEM63 family members.
DOI: https://doi.org/10.7554/eLife.41845.013
**Figure 2—figure supplement 2.** Protein sequence alignment of OSCA/TMEM63 family members (continued).
DOI: https://doi.org/10.7554/eLife.41845.014
**Figure 2—figure supplement 3.** Protein sequence alignment of OSCA/TMEM63 family members (continued).
DOI: https://doi.org/10.7554/eLife.41845.015
**Figure supplement 4.** Structural Features of OSCA1.2 and TMEM16 proteins.
DOI: https://doi.org/10.7554/eLife.41845.016

closely follow the topology and organization of mouse TMEM16A (mTMEM16A) Ca$^{2+}$-activated chloride channel (*Dang et al., 2017*; *Paulino et al., 2017a*; *Paulino et al., 2017b*) and *Nectria haematococca* TMEM16 (nhTMEM16), which is a Ca$^{2+}$-dependent lipid scramblase (*Brunner et al., 2014*) and nonselective ion channel (*Lee et al., 2016*) (*Figure 2C*; *Figure 2D*, *Figure 2—figure supplement 4A*; *Figure 2—figure supplement 4B*). Relative to these TMEM16 structures, OSCA1.2 has an additional N-terminal TM helix positioned at the periphery of the TMD (*Figure 2A*). To facilitate structural comparison between OSCA and TMEM16 proteins, we refer to this N-terminal TM helix as 'TM0', and the remaining TM helices 'TM1-TM10'. Though TM1-TM10 of OSCA1.2 and TMEM16A align well at the level of a single subunit, the relative orientation of one subunit to the other is

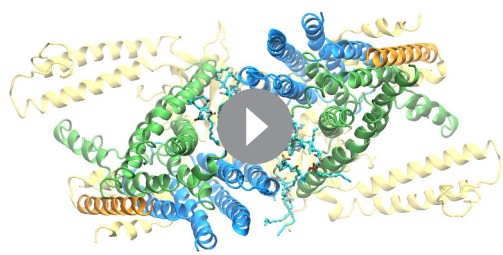

**Video 2.** Related to *Figure 1G–I*. The network of intersubunit lipids remains stable in the AT simulation. OSCA1.2 and POPC lipids in the atomistic simulation same as *Video 1*, showing the top view. Lipids found in the inter-subunit cleft are defined as within 20 Å of residue 655–683 (TM10) of both subunits, at the start of the AT simulation. Here the upper and lower leaflets contain 13 and 9 POPC molecules respectively. The networks of lipids in both leaflets are stable. No lipids either entered or left the cleft.

DOI: https://doi.org/10.7554/eLife.41845.011

different (Figure 2E). As a result of this distinct dimeric packing, the TMD of the OSCA1.2 dimer is more than 20 Å wider than mTMEM16A or nhTMEM16. Additional features distinguish the overall structure of OSCA1.2 from TMEM16 proteins. First, the intracellular domain of OSCA1.2 is mostly comprised of IL2, which connects TM2 and TM3, while the intracellular domains of mTMEM16a and nhTMEM16 are formed primarily by the N- and C- termini. Second, the regulatory $Ca^{2+}$ binding site composed of acidic and polar residues conserved across the TMEM16 family (*Brunner et al., 2014*) does not have a chemical environment conducive for $Ca^{2+}$ binding in OSCA1.2 (*Figure 2—figure supplement 4C*), consistent with distinct modes of regulation (mechanical activation vs. $Ca^{2+}$ dependence).

In the mTMEM16A dimer, there are two pores; one within each subunit (*Dang et al., 2017*; *Paulino et al., 2017a*). Topological similarities with mTMEM16A (*Figure 2*) suggest that OSCA1.2 also has two pores and could conduct ions through a structurally analogous pathway lined by TM3-TM7. The existence of two pores per OSCA1.2 dimer is consistent with the presence of a single subconductance state in stretch-activated single-channel currents (*Murthy et al., 2018*). To visualize the dimensions of the putative ion permeation pathway, we used the HOLE program (*Smart et al., 1996*) (*Figure 3A–C*, *Figure 3—figure supplement 1*, *Video 3*). Towards the extracellular side, this putative pore has an opening greater than 12 Å wide, which narrows into a 'neck' approximately 15 Å down the conduction pathway (*Figure 3A–C*). The neck extends for over 10 Å and reaches a minimum van der Waals radius of 0.5 Å, closing the channel. Mostly hydrophobic residues (I393, V396, A400, L434, L438, F471, V476, and F515) line the pore's constriction, as well as several charged and polar residues (Q397, S480, Y468, K512, and D523). Interestingly, a pair of π-helical turns (in TM5 and TM6a) are nearby the pore's neck region (*Figure 3C*). Because π-helices are energetically unstable, we speculate that π-to-α transitions at these positions could be associated with gating and, potentially, channel opening. In support of this idea, there are π-helical turns at highly similar locations of TM6 in mTMEM16A (*Paulino et al., 2017a*) and nhTMEM16 (*Brunner et al., 2014*), and a α-to-π transition in TM6 underlies $Ca^{2+}$-dependent activation in mTMEM16A (*Paulino et al., 2017a*) (*Figure 2—figure supplement 4D*). Similarly, the pore-lining helices of the epithelial calcium channel TRPV6 (*McGoldrick et al., 2018*) undergo π-to-α helix transitions during gating. Directly below the neck, the pore widens, and the side chains of E531, R572, and T568 create a hydrophilic environment that could stabilize hydrated ions (*Figure 3C*). Further below, the pore widens into membrane-exposed vestibule and is surrounded by IL2 and IL4 before exiting into the cytosol.

If TM3-TM7 indeed form the permeation pathway of OSCA1.2, mutation of residues that line the pathway should alter the channel's permeation or conductance properties. We chose to examine E531, which is the only pore-facing acidic residue conserved in this region across the OSCA/TMEM63 families and thus could contribute to conductance (*Figure 3C*, *Figure 2—figure supplements 1*, *2* and *3*, *Figure 2—figure supplement 4C*). We recorded and characterized stretch-activated currents in the cell-attached patch clamp mode from cells expressing wildtype or mutant (E531A) channels (*Figure 3D*). E531A had maximal current responses comparable to wildtype channels (*Figure 3D,E*), and modestly faster inactivation kinetics (*Figure 3E*). Strikingly, the mutation decreased the stretch-activated single-channel conductance by 1.6-fold, demonstrating that E531 contributes to the channel's ion permeation pathway (*Figure 3F,G*), and could play a role in binding or sequestering cations. Alternatively, the mutation could alter pore structure and dynamics owing to the interactions of the E531 side chain with R572 and Y605 (*Figure 3—figure supplement 1B*). Nonetheless, combining these mutagenesis data with structural homology to mTMEM16A

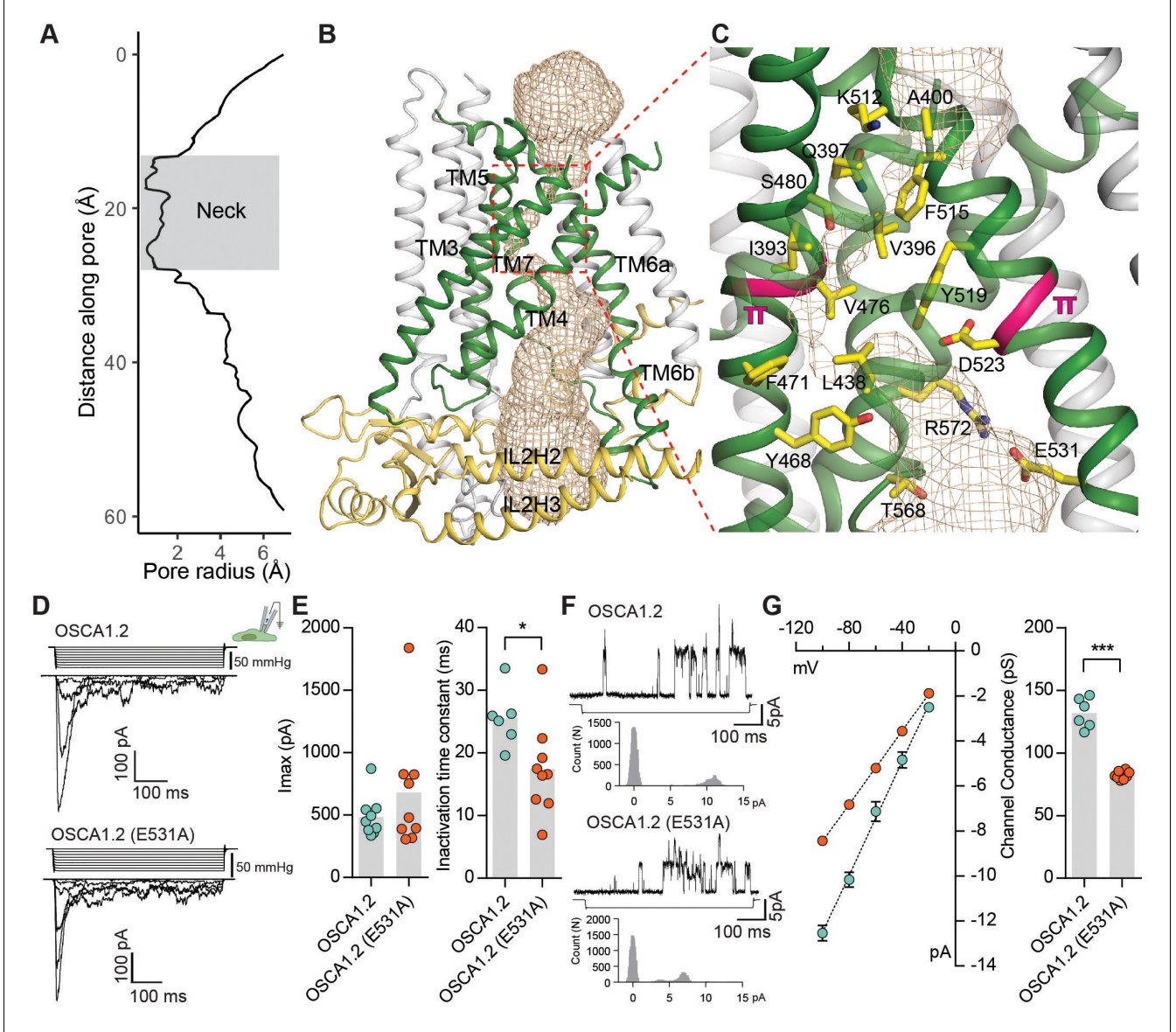

**Figure 3.** The pore of OSCA1.2. (**A**) Van der Waals radii of the pore plotted against axial distance. (**B**) Location of the pore in the context of the OSCA1.2 subunit. Pore-lining TMs are colored green, and pore pathway is depicted as wheat mesh. (**C**) Expanded view of the neck, and pore-lining residues. π-helical turns close to the neck are highlighted pink (also see *Video 4* and *Video 5*). (**D**) Representative traces of stretch-activated currents recorded from OSCA1.2- or OSCA1.2 (E531A)- expressing HEK-P1KO cells. In the cell-attached patch clamp mode, currents were elicited by applying negative pipette pressure in steps of Δ−10 mmHg. The corresponding stimulus trace is illustrated above the current trace. (**E**) Left, maximal current response from individual cells expressing OSCA1.2 (Imax: 487 ± 55 pA, N = 9) or OSCA1.2 (E531A) (Imax: 681 ± 161 pA, N = 9). Right, Inactivation time constant (ms) for individual cells across the two conditions. Bars represent population mean (OSCA1.2: 25.6 ± 2 ms (N = 6); OSCA1.2 (E531A): 17.4 ± 2 ms (N = 9). *p=0.01, Mann Whitney test). (**F**) Representative traces of stretch-activated single-channel currents (−80 mV) from wildtype or mutant channels. Channel openings are upward deflections. The negative pressure stimulus is illustrated below the current trace. Amplitude histogram for the representative current trace is represented below the stimulus trace. Single-channel amplitude was determined as the amplitude difference in Gaussian fits of the histogram. (**G**) Left, average I-V response curve for stretch-activated single-channel currents from wildtype or mutant channels. Right, channel conductance from individual cells across the two conditions. Bars represent population mean (OSCA1.2: 132 ± 5 ms (N = 6); OSCA1.2 (E531A): 82 ± 1 ms (N = 8). ***p=0.0007, Mann Whitney test).

DOI: https://doi.org/10.7554/eLife.41845.017

The following figure supplement is available for figure 3:

**Figure supplement 1.** Details of the OSCA 1.2 pore.
DOI: https://doi.org/10.7554/eLife.41845.018

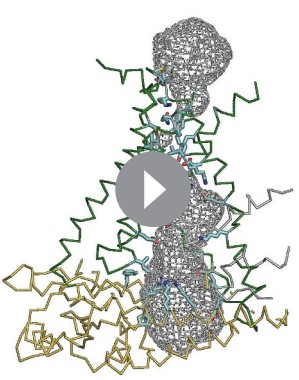

**Video 3.** View of OSCA1.2 pore from the membrane plane. Pore and pore lining residues. Side chains are shown as blue sticks, backbone of OSCA1.2 is shown in ribbon representation and colored green for the TM helices and yellow for the intracellular domain.
DOI: https://doi.org/10.7554/eLife.41845.019

(*Paulino et al., 2017a*; *Paulino et al., 2017b*), we conclude that TM3-TM7 most likely form the pore. This conclusion is further supported by MD simulations showing hydration of the pore pathway being consistent with the pore radius profile from HOLE (*Figure 1—figure supplement 5B*, *Figure 3—figure supplement 1C*, *Video 4*, *Video 5*).

## Discussion

Our structural study of OSCA1.2 elucidates a novel architecture for MA ion channel and reveals unanticipated similarities between OSCA1.2 and TMEM16 proteins, including pore structure. Which structural elements in OSCA1.2 could be involved in membrane tension-sensing and gating? Two notable features present in OSCA1.2, but not TMEM16s, are a hook-shaped loop that enters the membrane that intervenes IL2H2 and IL2H3 (*Figure 2A*, *Figure 4A*), and a horizontal amphipathic helix in IL1 (IL1H1) that slightly deforms the membrane lower leaflet in MD simulations (*Figure 4B*, *Video 1*). Their association with the cytoplasmic bilayer leaflet suggests

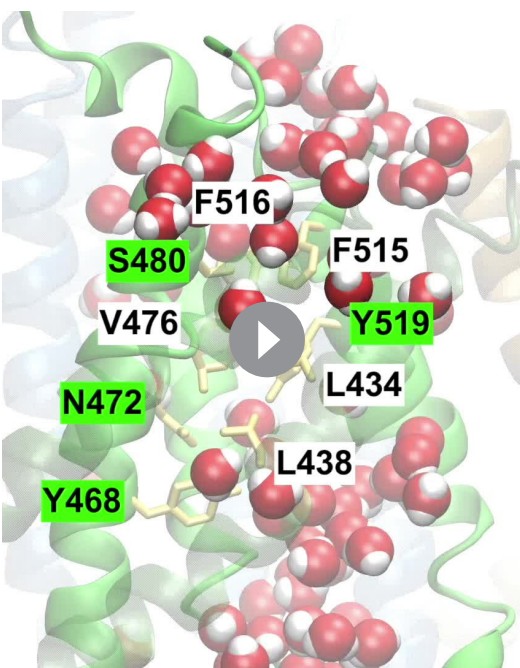

**Video 4.** Related to *Figure 3C* and *Figure 1—figure supplement 5B*. Water distribution around the neck region of the pore (subunit 1). Water molecules around the highlighted residues on TM4–6 are shown; residue labels are colored by residue types (green: polar; white: non-polar). In subunit 1, intermittent permeation of water between L434 (TM4) and Y519 (TM6) were observed when position restraints were removed after 50 ns. This water permeation pathway presented a possible alternative to that suggested by the HOLE program. See also *Video 5*.
DOI: https://doi.org/10.7554/eLife.41845.020

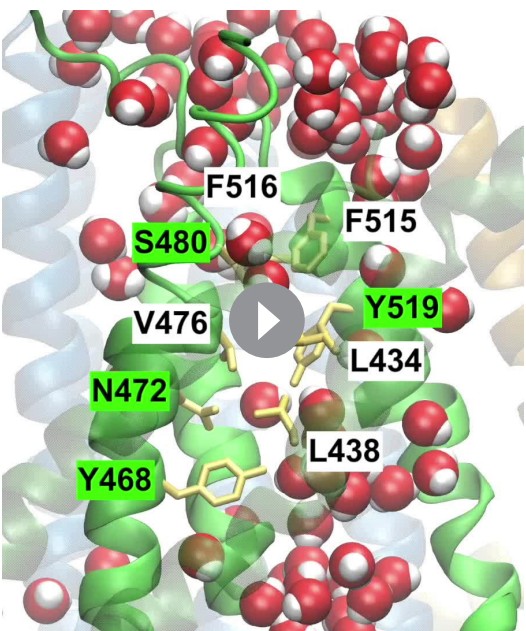

**Video 5.** Related to *Figure 3C* and *Figure 1—figure supplement 5B*. Water distribution around the neck region of the pore (subunit 2). Water molecules around the highlighted residues on TM4–6 are shown; residue labels are colored by residue types (green: polar; white: non-polar). Subunit 2 remains impermeable to water during the entire simulation, in contrast to that in subunit 1 (*Video 4*).
DOI: https://doi.org/10.7554/eLife.41845.021

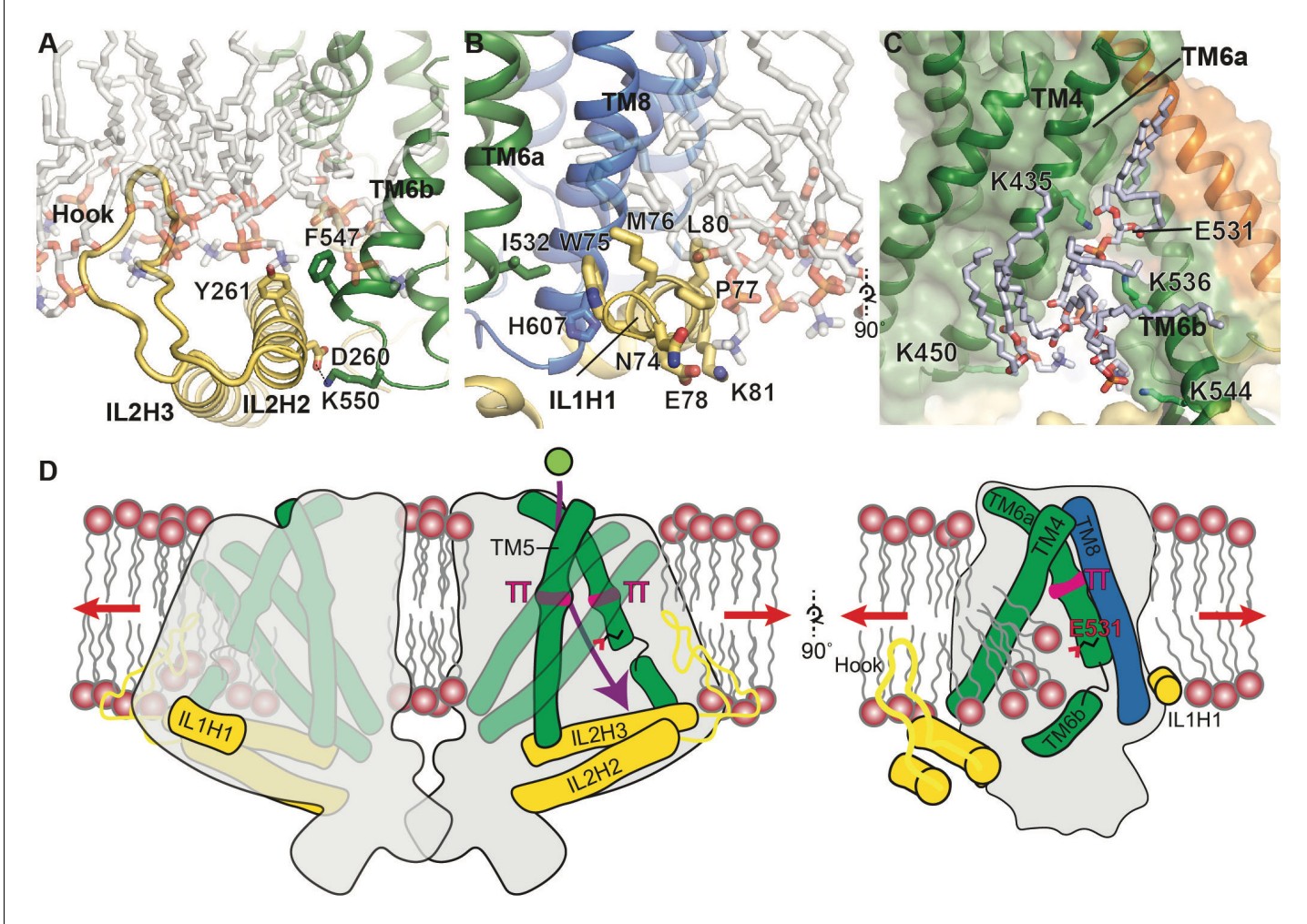

**Figure 4.** Putative mechanosensitive features of OSCA1.2. (**A**) Cartoon representation of the membrane embedded hook, which intervenes IL2H2 and IL2H3. IL2H2 interacts with pore-lining TM6b. (**B**) Cartoon and stick representation of the amphipathic membrane-embedded IL1H1, which contacts pore-lining TM6a and TM8. In (**A**) and (**B**), MD snapshots of lipids proximal to the hook and IL1H1 are shown, demonstrating that both domains are membrane-embedded. (**C**) A snapshot (at ~50 ns) of the MD simulation showing four phospholipids occupying a cytoplasmic cavity of the pore, which is formed by TM4 and TM6b. Interactions of three lipids with the sidechains of K435, K536, and K544 were sustained throughout the 1 μs of CG simulation and also throughout the entire 250 ns of AT simulation (also see *Figure 1—figure supplement 5B*, *Video 6* and *Video 7*). (**D**) Schematic showing various features of OSCA1.2 proposed to be involved in MA ion channel function. Red arrows depict direction of membrane tension when applied, though the current structure is not under tension. Green sphere represents ion entering the pore and purple arrow shows pore pathway that transits between π helices. The right panel highlights the amphipathic and membrane embedded regions of one monomer of OSCA1.2 and the local perturbation of the inner leaflet, including entry of lipids into the cytoplasmic side of the pore.

DOI: https://doi.org/10.7554/eLife.41845.022

these two features could be sensitive to membrane tension, akin to the N-terminal amphipathic helix in MscL (*Bavi et al., 2016*), and their movement could be coupled to channel conformation through their interactions with the TMD (*Figure 4A,B*). Another notable aspect of the OSCA1.2 structure, similar to TMEM16s, is that the lower region of the pore is exposed to the membrane and thus could permit lipid entry (*Figure 3B*, *Figure 3—figure supplement 1A,C*). Indeed, in MD simulations, lipids occupy and occlude the cytoplasmic half of the pore, with their phosphate head groups interacting with four lysine residues on TM4 and TM6b (*Figure 4C*, *Figure 1—figure supplement 5C*, *Video 6*, *Video 7*). It is possible that membrane tension promotes gating by affecting the lipid occupancy in this region, analogous to proposed mechanistic models for TRAAK (*Brohawn et al., 2014*) and MscS (*Pliotas et al., 2015*). Finally, a general principle of MA ion channels is that their cross-sectional area should increase in response to membrane tension (*Guo and MacKinnon, 2017*; *Haswell et al.,*

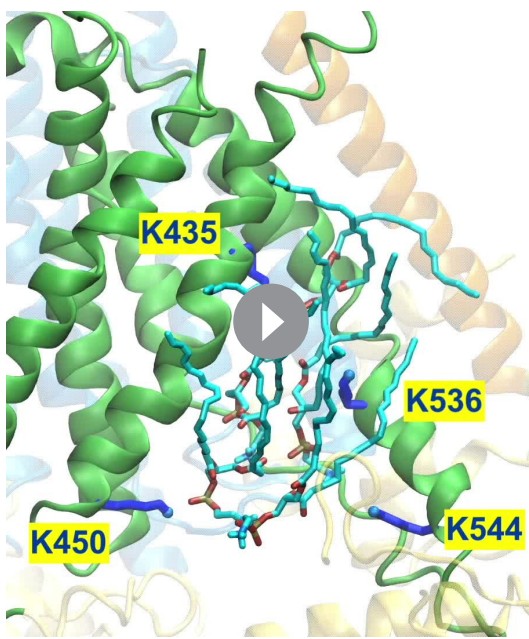

**Video 6.** Related to *Figure 4C* and *Figure 1—figure supplement 5C–D*. Lipid distribution around the cytoplasmic side of the pore (subunit 1).Interactions of lipid phosphate groups with lysine residues on TM4 and TM6b are shown (see also *Figure 1—figure supplement 5C* and *Figure 1—figure supplement 5D*). For both of the subunits, the orientation of K435 rearranges to face the constricted region of the pore after the position restraints are removed after 50 ns. The network of lipids of the lower leaflet shows how the bilayer is deformed around this region, with the lipid closest to K435 arranged almost perpendicular to the membrane (*Video 7*). As noted above (*Video 1*), lipid flip-flop is not observed.

DOI: https://doi.org/10.7554/eLife.41845.023

*place, 2016*), the shared TM topology between more distantly related TMEM16s and OSCA1.2 suggests a similar fold for TMCs. It is therefore likely that OSCA/TMEM63, TMEM16, and TMC all utilize a similar architectural scaffold to carry out various functions at the membrane. While the details underlying such functional diversity await discovery, our study establishes a framework to understand how OSCAs exploited this common fold, and evolved specific features, to serve the role of MA ion channels.

We note that cryo-EM structures of two other members of the OSCA/TMEM63 family, OSCA1.1 and OSCA3.1 (*Zhang et al., 2018*), were recently published after our initial BioRxiv deposition (*Jojoa-Cruz et al., 2018*).

*2011*; *Phillips et al., 2009*). Qualitatively, the absence of a dimeric protein-protein interface between membrane domains (*Figure 1I*) could allow OSCA1.2 to expand its cross-sectional area significantly in response to physiological lateral tension without the energetic penalty of breaking inter-subunit interactions within the membrane. A comparison of nanodisc-embedded and LMNG-solubilized structures hints at the positional flexibility of the subunits relative to each other while in the closed state (*Figure 1—figure supplement 4B,C*, *Video 8*). Larger movements will be expected under membrane tension and channel opening. The features of OSCA1.2 outlined above might act in concert to modulate channel gating in response to mechanical stimuli (*Figure 4D*).

Intriguingly, transmembrane-channel-like (TMC) proteins, which are pore-forming subunits of the hair cell mechanoelectrical transduction (MET) apparatus (*Fettiplace, 2016*; *Kawashima et al., 2011*; *Pan et al., 2013*), are distantly related to the TMEM16 family and proposed to have the same topology (*Ballesteros et al., 2018*; *Hahn et al., 2009*; *Medrano-Soto et al., 2018*; *Pan et al., 2018*). Although the molecular identity of the MET ion channel remains incompletely understood (*Fetti-*

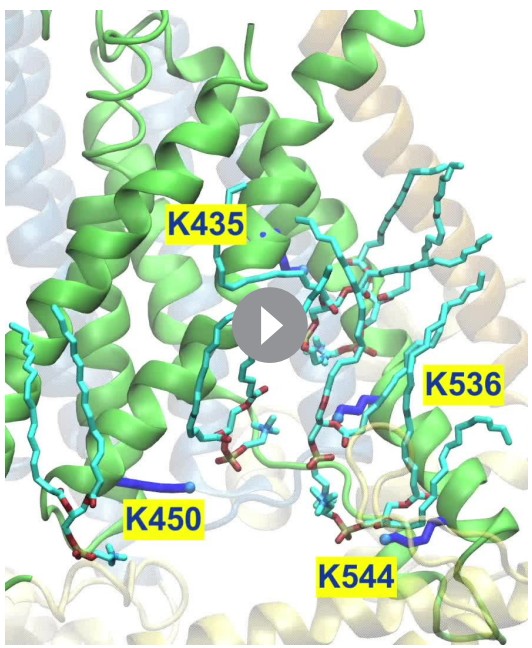

**Video 7.** Related to *Figure 4C* and *Figure 1—figure supplement 5C–D*. Lipid distribution around the cytoplasmic side of the pore (subunit 2). See *Video 6* for caption.

DOI: https://doi.org/10.7554/eLife.41845.024

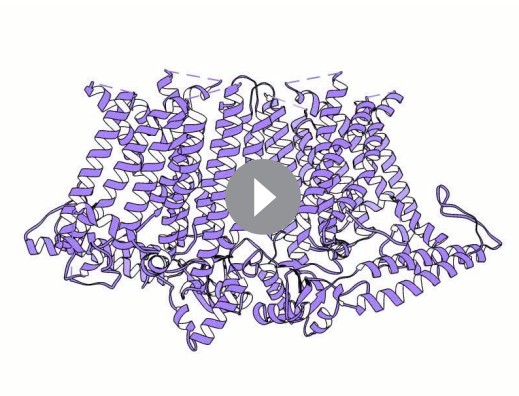

**Video 8.** Morph between LMNG-solubilized and nanodisc-embedded OSCA1.2 models.
DOI: https://doi.org/10.7554/eLife.41845.025

## Materials and methods

### Expression constructs

For structural studies, codon optimized OSCA1.2 gene (UniProt ID: Q5XEZ5) for expression in human cell lines was cloned into vector pcDNA3.1. An EGFP tag was placed at the C terminus and connected to the gene via a PreScission Protease cleavable linker (LEVLFQGP). A FLAG tag (DYKDDDDK) was added to the C terminus of EGFP with two intervening alanines as a linker. We refer to this construct as OSCA1.2-PP-EGFP.

### Protein expression and purification

To obtain samples of OSCA1.2 solubilized in LMNG, four liters of HEK293F cells (ThermoFisher Freestyle 293 F, RRID: CVCL_D603) were grown in Freestyle 293 expression media to a density of $1.2–1.7 \times 10^6$ cells/mL. All cell lines tested negative for mycoplasma contamination. Each liter was transfected by combining 1 mg/L of the construct with 3 mg/L of PEI MAX 40K in 30 mL of Opti-MEM and then adding the mix to the culture of cells. Transfected cells were grown for 48 hr and then pelleted, washed with ice cold PBS, flash frozen and stored at −80°C for future use. From this point forward, every step of the purification was carried out at 4°C unless otherwise stated. Pellets were thawed on ice, resuspended in 200 mL of solubilization buffer (25 mM tris pH 8.0, 150 mM NaCl, 1% LMNG, 0.1% CHS, 2 µg/mL leupeptin, 2 µg/mL aprotinin, 1 mM phenylmethylsulfonyl fluoride (PMSF), 2 µM pepstatin, 2 mM DTT) and stirred vigorously for 2–3 hr. Subsequently, insoluble material was pelleted via ultracentrifugation for 45 min at 92,387 $g$ in a Type 70 Ti rotor. Batch binding of the supernatant was performed for 1 hr with 2 mL of GFP nanobody (*Kirchhofer et al., 2010*)-coupled CNBr-Activated Sepharose 4B resin that had been previously equilibrated with wash buffer (25 mM tris pH 8.0, 150 mM NaCl, 0.01% LMNG, 0.001% CHS, 2 mM DTT). Protein bound to resin was spun down for 2 min at low speed and washed with wash buffer twice and transferred to a gravity flow column. Resin was wash with 10 CV of wash buffer and transferred to a conical tube using 5 to 8 mL of wash buffer, followed by addition of ~15 µg of PreScission Protease and rotated overnight. Both resin and supernatant were transferred into a gravity flow column and the flow through was collected and concentrated using a 100 kDa MWCO Amicon Ultra centrifugal filter. Concentrated protein was injected onto Shimadzu HPLC and SEC was performed using a Superose 6 Increase column equilibrated to wash buffer. Fractions corresponding to OSCA1.2 were concentrated to 4.4 mg/mL. Typical yields of OSCA1.2 using this procedure were ~60 µg purified product per liter of HEK293F cells.

To obtain samples of nanodisc-embedded OSCA1.2, the same purification procedures were used with minor changes. Instead of LMNG, solubilization buffer and wash buffer had 1%/0.1% and 0.01%/0.001% n-dodecyl beta-D-maltopyranoside (DDM)/CHS, respectively. After SEC, protein was concentrated down to approximately 200 µL and mixed with MSP2N2 and soybean polar lipid extract (Avanti #541602) at a molar ratio of 1:3:166 (protein monomer:MSP2N2:lipid). Mixture was rocked at 4°C for one hour. 10 mg of Bio-beads SM2 (Bio-rad) prewashed with SEC2 buffer (20 mM Tris pH 8.0, 150 mM NaCl, 1 mM DTT) were added to the mixture and rotated at 4°C. After one hour, another 10 mg of prewashed biobeads were added and rotated overnight at 4°C. Biobeads were removed and protein concentrated and injected onto Shimadzu HPLC for a second SEC using SEC2 buffer and Superose 6 Increase column. Fractions corresponding to OSCA1.2 peak were concentrated to 2.1 mg/mL. Typical yields of OSCA1.2 in nanodiscs using this procedure were ~30 µg purified product per liter of HEK293F cells.

## Cryo-EM sample preparation and data collection

3.5 µL of purified protein (at 4.4 mg/mL for LMNG-solubilized OSCA1.2 and 2.1 mg/mL for nano-disc-embedded OSCA1.2) was applied to previously plasma-cleaned UltrAuFoil 1.2/1.3 300 mesh grids and blotted once for 3.5 s with blot force 0 after a wait time of 12 s. Blotted grids were plunge frozen into nitrogen-cooled liquid ethane using a Vitrobot Mark IV (ThermoFisher). Images for LMNG-solubilized OSCA1.2 were collected at 300 kV using a Titan Krios (ThermoFisher) coupled with a K2 Summit direct electron detector (Gatan) at a nominal magnification of 29,000x with a pixel size of 1.03 Å. 51 frames were collected per movie for a total accumulated dose of ~60 electrons per Å$^2$ using a defocus range of −1.0 to −2.6 µm. For nanodisc-embedded OSCA1.2, a Talos Arctica at 200 kV was used with a K2 Summit direct electron detector at a nominal magnification of 36000x with 1.15 Å pixel size. 54 frames were collected per movie for a total accumulated dose of ~60 electrons per Å$^2$ using a defocus range of −0.4 to −2.2 µm. Automated micrograph collection was performed using Leginon software (*Suloway et al., 2005*), collecting 2536 and 1740 total movies for LMNG-solubilized and nanodisc-embedded OSCA1.2, respectively. Movies were aligned and dose-weighted using MotionCor2 (*Zheng et al., 2017*).

## Cryo-EM image processing

Image assessment and masking of dose-weighted micrographs was performed using EMHP (*Berndsen et al., 2017*) and CTF values were obtained with Gctf on non-dose-weighted micrographs (*Zhang, 2016*). A small subset of micrographs at a range of defocus values were used for manual picking and 2D classification in RELION-2.1 (*Kimanius et al., 2016*) to generate templates for the dataset. For LMNG-solubilized OSCA1.2, template picking on 1357 assessed micrographs was done in RELION-2.1 and followed by particle extraction. Particles were imported into cryoSPARC v0.6.5 (*Punjani et al., 2017*). For downstream image processing, all parameters in cryoSPARC were kept as default unless otherwise stated. 326,398 particles were subjected to *ab-initio* reconstruction requesting three classes. Homogeneous refinement with C2 symmetry imposed was performed using the particles and output structure of the best class (low pass filtered to 30 Å) as reference. Particles from the best two classes from the initial *ab-initio* reconstruction run were then subjected to a second round of *ab-initio* reconstruction, this time requesting four classes. The best two classes (134,337 particles) were refined with C2 symmetry using the initial model obtained from the previous refinement, resulting in the final 3.5 Å resolution map. For nanodisc-embedded OSCA1.2, templates were generated as previously described, and particles were picked and extracted in RELION-2.1 on 1211 micrographs. 788,948 template-picked particles were imported into cryoSPARC. One round of 2D classification was performed and best 2D classes were selected (675,536 particles). Two consecutive rounds of *ab-initio* reconstruction requesting three classes were performed, and particles belonging to the best class were taken for further processing. 92,280 particles from the best class from the last round *ab-initio* reconstruction were subjected to homogeneous refinement with C2 symmetry imposed using the final LMNG solubilized map low pass filtered 30 Å as an initial model, resulting in a 3.2 Å resolution map. To improve quality of the map, particles from micrographs with an estimated resolution (calculated by Gctf) worse than 5 Å were excluded from the dataset. The remaining 91,729 particles were re-extracted and imported into cryoSPARC, followed by 2D classification. To obtain an initial model based on this dataset, C2 symmetry imposed homogeneous refinement was performed on particles from the best 2D classes (91,729 particles) using the LMNG-solubilized OSCA1.2 structure low pass filtered to 30 Å as the reference model. Subsequently, this model and its particles were used for heterogeneous refinement (three classes). Best 3D classes were pooled together and refined, resulting in a 3.2 Å map (76,797 particles). Particles for this map were exported from cryoSPARC and per particle CTF was estimated (*Zhang, 2016*). A first round of masked refinement was performed in RELION-2.1 using the map generated in cryoSPARC (low-pass filtered to 40 Å) as a reference. The output map from this refinement was then used to create a new mask, which was used for a second round of 3D refinement in RELION-2.1, resulting in a 3.1 Å resolution map. Using the parameter –solvent_correct_fsc was useful for obtaining a higher resolution structure.

## Model building and refinement

Manual model building of OSCA1.2 was carried out in coot (*Emsley and Cowtan, 2004*), iterated with real space refinement using phenix (*Adams et al., 2010*) and Rosetta (*Wang et al., 2016*). The atomic model was first built and refined against the 3.1 Å resolution map of nanodisc-embedded OSCA1.2 sharpened to a b-factor of $-81$ Å$^2$ automatically determined by RELION-2.1 postprocessing (*Kimanius et al., 2016*). Registry was initially aided by loose homology of TM4, TM5 and TM6 to mouse TMEM16A (*Paulino et al., 2017a*) (PDB ID: 5OYB). The refined structure of nanodisc-embedded OSCA1.2 was then docked into the 3.5 Å resolution map of LMNG-solubilized OSCA1.2, sharpened to a b-factor of $-170$ Å$^2$ in cryoSPARC, then manually adjusted and re-refined. For real space refinement using Rosetta (*Wang et al., 2016*), the Relax protocol was used, requesting an output of 157 refined structural models. The top scoring structure was chosen after evaluation with MolProbity (*Chen et al., 2010*), EMRinger (*Barad et al., 2015*), and manual inspection. In each structure, the model includes residues 3–50, 71–122, 156–401, 420–490, 502–717, totaling 634 of 771 residues in the full-length sequence of OSCA1.2. Structures were validated by MolProbity (*Chen et al., 2010*), EMRinger (*Barad et al., 2015*), and by computing map-to-model FSC plots using phenix.mtriage (*Afonine et al., 2018*). Validations were completed with all side chains in the model. However, side chains of residues 268–287 in the membrane hook were truncated to ß-carbon in the deposited models because of relatively poor density in this region. Structural figures were made in PyMOL (*Schrodinger, 2015*) or UCSF Chimera (*Pettersen et al., 2004*). Pore dimensions were calculated using HOLE (*Smart et al., 1996*). Alignment of the amino acid sequences was done using Clustal Omega (*Sievers et al., 2011*) and represented using ESPript 3.0 (*Robert and Gouet, 2014*).

## Molecular dynamics simulations

Molecular dynamics simulations were performed using GROMACS 5.0.2 (*Abraham et al., 2015*) (www.gromacs.org) based on the nanodisc-embedded structure. MemProtMD (*Stansfeld et al., 2015*) was used to convert the structure into a coarse-grained (CG) representation using the MARTINI 2.2 (*de Jong et al., 2013*) force field, which was then embedded in a band of randomly oriented POPC molecules. Water and 0.15 M NaCl were then added to the periodic simulation box. Initial 100 ns simulation with protein backbone beads position restrained permitted lipid self-assembly (force constant 1000 kJ mol$^{-1}$ nm$^{-2}$). The position restraints were then removed for five runs of a 1-μs equilibrium simulation, during which a simple elastic network based on the standard Martini topology (*Periole et al., 2009*) with a force constant 1000 kJ mol$^{-1}$ nm$^{-2}$ was applied to the protein (lower and upper cut-off were 5 and 9 Å). Protein-lipid contact analysis was performed with a locally written script that employed a 6 Å cut-off, based on the last 800 ns of the simulation. Final contact values were reported as an average of the two subunits in the five runs. We then converted the final frame of one 1 μs CG simulation to atomistic (AT) detail using a fragment-based protocol CG2AT-Align (*Stansfeld and Sansom, 2011*) (force field: OPLS united atom (*Robertson et al., 2015*)), solvated again in TIP4P water (*Neumann, 1986*) with 150 mM NaCl. We then performed a 250 ns equilibrium simulation with protein non-hydrogen atoms position restrained (force constant as above) in the initial 50 ns only. All CG and AT were performed as NPT ensembles held at 310 K (except CG lipid assembly at 323 K) and 1 bar. The time-step for CG and AT simulations were 20 and 2 fs respectively. For the AT simulation, a velocity-rescaling thermostat (*Bussi et al., 2007*) with a coupling constant of 0.1 ps, and a semi-isotropic Parrinello-Rahman barostat$^R$ with a coupling constant of 1 ps and a compressibility of $4.5 \times 10^{-5}$ bar$^{-1}$ were used. The LINCS algorithm (*Hess, 2008*) was used to constrain all covalent bonds. Electrostatics were modelled using a Particle Mesh Ewald (PME) model (*Essmann et al., 1995*) whereas van der Waals' interactions were modelled using a cut-off scheme, both with cut-off at 1 nm. Simulations figures were created with VMD (*Humphrey et al., 1996*) and PyMOL (*Schrodinger, 2015*). All figures of simulations represent the final snapshot of the AT simulation unless otherwise stated.

## Generation of mutant, cell culture and transfections

The E531A substitution in OSCA1.2 was generated using Q5 Site-Directed Mutagenesis Kit (New England BioLabs) according to the manufacturer's instruction and confirmed by full-length DNA sequencing. OSCA1.2-PP-EGFP and OSCA1.2(E531A)-PP-EGFP were transfected and tested in Piezo1-knockout (P1KO) HEK293T cells. HEK293T-P1KO cells were generated in house using

CRISPR–Cas9 nuclease genome editing technique as described previously (*Dubin et al., 2017*; *Lukacs et al., 2015*), and were negative for mycoplasma contamination. Cells were grown in DMEM containing 4.5 mg mL$^{-1}$ glucose, 10% fetal bovine serum, 50 U mL$^{-1}$ penicillin and 50 µg mL$^{-1}$ streptomycin. Cells were plated onto 12 mm round glass poly-D-lysine coated coverslips placed in 24-well plates and transfected using Lipofectamine 2000 (Invitrogen) according to the manufacturer's instruction. All plasmids were transfected at a concentration of 600 ng mL$^{-1}$. Cells were recorded from 24 to 48 hr after transfection. The PP-EGFP fusion at the C-terminus of the protein did not affect channel expression or MA current properties.

## Electrophysiology

Stretch-activated currents were recorded in the cell-attached patch-clamp configuration using an Axopatch 200B amplifier (Axon Instruments). Currents were filtered at 2 kHz and sampled at 20 kHz. Leak currents before mechanical stimulations were subtracted off-line from the current traces. Membrane patches were stimulated with a 500 ms negative pressure pulse through the recording pipette using Clampex-controlled pressure clamp HSPC-1 device (ALA scientific). Since the single-channel amplitude is independent of the pressure intensity, the most optimal pressure stimulation was used to elicit responses that allowed single-channel amplitude measurements. These stimulation values were largely dependent on the number of channels in a given patch of the recording cell. Single-channel amplitude at a given potential was measured from trace histograms of 5–10 repeated recordings. Histograms were fitted with Gaussian equations using Clampfit 10.6 software. Single-channel slope conductance for each individual cell was calculated from linear regression curve fit to single-channel *I–V* plots.

For cell-attached patch-clamp recordings, external solution used to zero the membrane potential consisted of (in mM) 140 KCl, 1 MgCl$_2$, 10 glucose and 10 HEPES (pH 7.3 with KOH). Recording pipettes were of 1–3 MΩ resistance when filled with standard solution composed of (in mM) 130 mM NaCl, 5 KCl, 1 CaCl$_2$, 1 MgCl$_2$, 10 TEA-Cl and 10 HEPES (pH 7.3 with NaOH).

## Data and materials availability

Cryo-EM maps of OSCA1.2 in nanodiscs and LMNG have been deposited to the Electron Microscopy Data Bank under accession codes 9112 and 9113. Atomic coordinates of OSCA1.2 in nanodiscs and LMNG have been deposited to the PDB under IDs 6MGV and 6MGW. All other data are available upon request to the corresponding author(s).

## Acknowledgments

 We thank W. Anderson for managing the electron microscopy facility at Scripps Research, H Turner for helping with data collection, and C Bowman for assistance with computation. We acknowledge J Kefauver, R Kirchdoerfer, and members of the Ward lab for helpful advice. KS thanks VK and DD for discussion.

This work was supported by a Ray Thomas Edwards Foundation grant to ABW, and NINDS grant 1R35NS105067 to AP. Work in MSPS's lab is supported by Wellcome (grant 208361/Z/17/Z), BBSRC (grants BB/N000145/1 and BB/R00126X/1), and EPSRC (grant EP/R004722/1). KS is a postdoctoral fellow of the Jane Coffin Childs Memorial Fund for Medical Research. CCAT is supported by the Skaggs-Oxford Scholarship and the Croucher Foundation. AP is an investigator of the Howard Hughes Medical Institute.

## Additional information

### Funding

| Funder | Grant reference number | Author |
| --- | --- | --- |
| Howard Hughes Medical Institute | | Ardem Patapoutian |
| National Institute of Neurological Disorders and Stroke | 1R35NS105067 | Ardem Patapoutian |

| Ray Thomas Edwards Foundation | | Andrew B Ward |
|---|---|---|
| Wellcome | 208361/Z/17/Z | Mark SP Sansom |
| Biotechnology and Biological Sciences Research Council | BB/N000145/1 | Mark SP Sansom |
| Biotechnology and Biological Sciences Research Council | BB/R00126X/1 | Mark SP Sansom |
| Engineering and Physical Sciences Research Council | EP/R004722/1 | Mark SP Sansom |
| Jane Coffin Childs Memorial Fund for Medical Research | | Kei Saotome |
| Skaggs-Oxford Scholarship | | Che Chun (Alex) Tsui |
| Croucher Foundation | | Che Chun (Alex) Tsui |

The funders had no role in study design, data collection and interpretation, or the decision to submit the work for publication.

### Author contributions

Sebastian Jojoa-Cruz, Conceptualization, Formal analysis, Investigation, Visualization, Methodology, Writing—original draft, Writing—review and editing; Kei Saotome, Conceptualization, Formal analysis, Funding acquisition, Investigation, Visualization, Methodology, Writing—original draft, Writing—review and editing; Swetha E Murthy, Conceptualization, Formal analysis, Investigation, Visualization, Methodology, Writing—review and editing; Che Chun Alex Tsui, Formal analysis, Funding acquisition, Investigation, Visualization, Methodology, Writing—review and editing; Mark SP Sansom, Resources, Supervision, Funding acquisition, Project administration, Writing—review and editing; Ardem Patapoutian, Andrew B Ward, Conceptualization, Resources, Supervision, Funding acquisition, Project administration, Writing—review and editing

### Author ORCIDs

Sebastian Jojoa-Cruz (iD) http://orcid.org/0000-0002-4392-3898
Kei Saotome (iD) http://orcid.org/0000-0002-4135-5356
Swetha E Murthy (iD) http://orcid.org/0000-0001-9580-3380
Che Chun Alex Tsui (iD) http://orcid.org/0000-0003-4886-9824
Mark SP Sansom (iD) http://orcid.org/0000-0001-6360-7959
Ardem Patapoutian (iD) http://orcid.org/0000-0003-0726-7034
Andrew B Ward (iD) https://orcid.org/0000-0001-7153-3769

### Decision letter and Author response

Decision letter https://doi.org/10.7554/eLife.41845.036
Author response https://doi.org/10.7554/eLife.41845.037

## Additional files

### Supplementary files

• Transparent reporting form
DOI: https://doi.org/10.7554/eLife.41845.026

### Data availability

Cryo-EM maps of OSCA1.2 in nanodiscs and LMNG have been deposited to the Electron Microscopy Data Bank under accession codes 9112 and 9113. Atomic coordinates of OSCA1.2 in nanodiscs and LMNG have been deposited to the PDB under IDs 6MGV and 6MGW. Due to their large size (300Gb+), the raw data files are available upon request to the corresponding author(s).

The following datasets were generated:

| | | Database and |
|---|---|---|

| Author(s) | Year | Dataset title | Dataset URL | Identifier |
|---|---|---|---|---|
| Jojoa-Cruz S, Saotome K, Patapoutian A, Ward AB | 2018 | Cryo-EM map of mechanically activated ion channel OSCA1.2 in nanodisc | https://www.ebi.ac.uk/pdbe/entry/emdb/EMD-9112 | Electron Microscopy Data Bank, 9112 |
| Jojoa-Cruz S, Saotome K, Patapoutian A, Ward AB | 2018 | Cryo-EM map of mechanically activated ion channel OSCA1.2 in LMNG | https://www.ebi.ac.uk/pdbe/entry/emdb/EMD-9113 | Electron Microscopy Data Bank, 9113 |
| Jojoa-Cruz S, Saotome K, Patapoutian A, Ward AB | 2018 | Structure of mechanically activated ion channel OSCA1.2 in nanodisc | https://www.rcsb.org/structure/6MGV | RCSB Protein Data Bank, 6MGV |
| Jojoa-Cruz S, Saotome K, Patapoutian A, Ward AB, Jojoa-Cruz S | 2018 | Structure of mechanically activated ion channel OSCA1.2 in LMNG | https://www.rcsb.org/structure/6MGW | RCSB Protein Data Bank, 6MGW |

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
