## [Decision Letter]

Congratulations, we are pleased to inform you that your article, "Cryo-EM structure of the mechanically activated ion channel OSCA1.2", has been accepted for publication in *eLife*.

In their manuscript, the authors describe the structure of the mechanically activated channel OSCA1.2 determined by cryo-EM in detergent and lipid nanodiscs. The data is of very high quality and both structures appear to be well refined. Whereas the bulk of the functional data supporting the conclusions are presented in an accompanying manuscript, the work also includes the functional characterization of a putative pore-lining mutant and data from molecular dynamics simulations that both support the putative location of the pore. Although this pore is discontinuous, as expected for a stable closed conformation in the absence of force, the proposal is credible. In summary, I think that this is a very exciting study, which describes a novel architecture of a mechanically activated membrane protein. The recently published structure of a homolog does in no way compromise the general interest of this work.

Minor comments are listed below:

1) Please specify which PEI was used in large-scale transfections, in Materials and methods.

2) There are a couple of Ramachandran outliers that should be fixed (one lysine per chain in the membrane hook).

3) Please provide a bit more detail (in the Materials and methods) on how Rosetta was used in model refinement.

4) Figure 1—figure supplement 1A, H: is the y-axis trp fluorescence or A280?

5) Figure 1—figure supplement 1C, J: were the apparently junk classes included in refinement? For example, panel J, row 3, column 1?

6) Figure 1—figure supplement 2: at the final step, can you please elaborate a bit more on what is meant by Relion 3D autorefine x2? Detail added in the legend or in the Materials and methods would be fine.

7) I am generally curious about confidence in positioning of anionic side chains proposed to be important in conformational transitions or permeation. Specifically, please provide a figure panel or supplement panel that shows EM density quality for side chains: Figure 2—figure supplement 2C, in particular the E531 and its electrostatic interactions; D523 vicinity in Figure 3C. If density for these residues is clearly displayed already in Figure 1—figure supplement 3, please just indicate them with an arrow or asterisk. The equivalent residue to E531 appears to have been modeled quite differently in the recent NSMB paper (Zhang et al., 2018) on OSCA channels – it may be that the density is simply not clear in one or both cases.

8) It should be mentioned whether the data shows bound lipids in the vicinity of the protein, particularly at the region between the two subunits. It would also be interesting to learn more about features of the lipid environment surrounding the dimeric protein from a low-pass filtered map of the protein in nanodiscs. Is there any evidence that protein induces distortions of the membrane environment as observed in MD simulations?

9) Since the MD simulations are an important part of the work, they might deserve better coverage in the Results. It can be expected that the gap between subunits would be filled with lipids and the wide entrances to the pore region would be occupied by water. Is this not already the case in the initial setups used for simulation? Was the equilibration of the system the only purpose of the atomistic simulations? In Figure 1—figure supplement 6, the authors show snapshot after 50 ns of simulations. It would be interesting to know what has changed in the solvent, protein and membrane region compared to the starting structure.

10) The authors speculate that gap between two helices at the intracellular part of the pore leading to its exposure to the membrane might play a role during channel activation. It could be mentioned that a similar opening is found in TMEM16A, which has so far not been described to be mechanically gated.

11) The authors emphasize the similarity between detergent and nanodisc structures in the Results but propose that differences between both structures might indicate their movements during activation in the Discussion. This is confusing. I suggest that the small differences between both structures could already be better described in the Results.

12) Results, fifth paragraph: Ca^2+^-activation in TMEM16A involves an α-to-π transition.

13) It would be helpful to show Figure 3—figure supplement 1A as stereo figure.

---

## [Author Response]

Minor comments are listed below:1) Please specify which PEI was used in large-scale transfections, in Materials and methods.

PEI MAX 40K was used for large scale transfections. This detail has been added in the Materials and methods section.

2) There are a couple of Ramachandran outliers that should be fixed (one lysine per chain in the membrane hook).

We have decided not to fix these outliers due to this region being flexible and the lowest resolution region of our maps. Our current models represent our best attempt at fitting the residues of the membrane hook inside the density, which resulted in one outlier in this region.

3) Please provide a bit more detail (in the Materials and methods) on how Rosetta was used in model refinement.

More detail has been added in the Materials and methods section.

4) Figure 1—figure supplement 1A, H: is the y-axis trp fluorescence or A280?

Changes have been made on Figure 1—figure supplement 1 to show that the measurement corresponds to UV absorbance (A280).

5) Figure 1—figure supplement 1C, J: were the apparently junk classes included in refinement? For example, panel J, row 3, column 1?

For LMNG-solubilized OSCA1.2 (Figure 1—figure supplement 1J), no filtering of particles was done in the 2D classification stage. Data were processed with and without filtering for the best 2D classes. The final reconstruction did not differ greatly between both alternatives. Furthermore, a 2D classification of the final 3.5 Å model shows that particles that made up the junk classes were filtered out during *ab-initio* reconstruction. For nanodisc-embedded OSCA1.2 structure (Figure 1—figure supplement 1C), 2D classification was performed using cryoSPARC and only the best classes were used for subsequent analysis, as explained in the Materials and methods.

6) Figure 1—figure supplement 2: at the final step, can you please elaborate a bit more on what is meant by Relion 3D autorefine x2? Detail added in the legend or in the Materials and methods would be fine.

“Relion 3D autorefine x2” means two subsequent rounds of 3D autorefine using relion. The second round uses the results of the previous one as input. This has been made clearer in the Materials and methods section.

7) I am generally curious about confidence in positioning of anionic side chains proposed to be important in conformational transitions or permeation. Specifically, please provide a figure panel or supplement panel that shows EM density quality for side chains: Figure 2—figure supplement 2C, in particular the E531 and its electrostatic interactions; D523 vicinity in Figure 3C. If density for these residues is clearly displayed already in Figure 1—figure supplement 3, please just indicate them with an arrow or asterisk. The equivalent residue to E531 appears to have been modeled quite differently in the recent NSMB paper (Zhang et al., 2018) on OSCA channels – it may be that the density is simply not clear in one or both cases.

As requested, residues D523 and E531 have been labeled in Figure 1—figure supplement 3. Although the density for E531 only reaches the β-carbon and it is possible that the side chain has a different rotamer, the fact that E531 is facing the pore remains the same.

8) It should be mentioned whether the data shows bound lipids in the vicinity of the protein, particularly at the region between the two subunits. It would also be interesting to learn more about features of the lipid environment surrounding the dimeric protein from a low-pass filtered map of the protein in nanodiscs. Is there any evidence that protein induces distortions of the membrane environment as observed in MD simulations?

There are in fact small densities in the inter-subunit cleft. Although these densities could correspond to lipids or cholesterol, there is also the chance that they represent detergent carried over from the purification or refined noise during processing. Thus, we have decided not to explicitly assign these densities as bound lipids. Regarding the lipid environment, in our current nanodisc map we do not observe clear evidence for the protein causing distortion of the membrane environment.

9) Since the MD simulations are an important part of the work, they might deserve better coverage in the Results. It can be expected that the gap between subunits would be filled with lipids and the wide entrances to the pore region would be occupied by water. Is this not already the case in the initial setups used for simulation? Was the equilibration of the system the only purpose of the atomistic simulations? In Figure 1—figure supplement 6, the authors show snapshot after 50 ns of simulations. It would be interesting to know what has changed in the solvent, protein and membrane region compared to the starting structure.

Videos are now added to reflect the arrangement of lipid and water molecules during the simulation. Indeed, the initial lipid self-assembly already placed lipids in the inter-subunit cleft, and waters occupying the entrance into the pore. While the equilibration of the system is one of the purposes of the atomistic simulation, this also allowed us to probe the dynamic interactions of waters and lipids with the protein. To this end, we have extended the original AT simulation for an additional 200 ns, without any restraints. This allows us to observe how the mechanosensing features of OSCA1.2 evolve when the system is allowed to relax. The changes of the protein, lipids and waters in the pore are now displayed in the videos and specific changes/lack of changes are described in the accompanying captions.

10) The authors speculate that gap between two helices at the intracellular part of the pore leading to its exposure to the membrane might play a role during channel activation. It could be mentioned that a similar opening is found in TMEM16A, which has so far not been described to be mechanically gated.

We have added in the text that the opening is similar to the one found in TMEM16 structures.

11) The authors emphasize the similarity between detergent and nanodisc structures in the Results but propose that differences between both structures might indicate their movements during activation in the Discussion. This is confusing. I suggest that the small differences between both structures could already be better described in the Results.

Our initial statement compared the LMNG-solubilized and nanodisc-embedded OSCA1.2 structures to highlight which regions may be more flexible in the closed state. However, we do not claim that these movements are the ones that lead to activation of the channel. We have slightly modified the statement in hopes to make it clearer. Given that the changes between the two structures are better represented on Video 8 than what we could do in words, we have decided not to further discuss it in the text.

12) Results, fifth paragraph: Ca^2+^-activation in TMEM16A involves an α-to-π transition.

The reviewer is correct. This has been corrected in the text. We thank the reviewer for catching this mistake.

13) It would be helpful to show Figure 3—figure supplement 1A as stereo figure.

In order to incorporate the reviewer’s suggestion, but also make this view accessible to everyone (not everyone is trained in seeing stereo views), we have decided to keep Figure 3—figure supplement 1 as is and we have added Video 3, which shows a 360° view of the pore.